# Investigations into Nonlinear Effects of Normal Pressures on Dynamic Cyclic Responses of Novel 3D-Printed TPMS Bridge Bearings

Pasakorn Sengsri and Sakdirat Kaewunruen *

Laboratory for Track Engineering and Operations for Future Uncertainties (TOFU Lab), School of Engineering, University of Birmingham, Edgbaston, Birmingham B15 2TT, UK
* Correspondence: s.kaewunruen@bham.ac.uk

**Abstract:** Bridge bearings are one of the most important components in bridge systems. Typical bearings are extensively used in small- to medium-span highway bridges since they are economical and offer a good performance at service-level conditions. On the other hand, common bridge bearings possess a low performance-to-weight ratio under combined compression and shear loading conditions (low crashworthiness and specific energy absorption), due to their heavy weight, high costs, and the non-recyclability of steel and elastomer materials. With the help of a relatively higher ratio of a 3D-printed triply periodic minimal surface (TPMS) structure, this method can potentially be used for bridge bearing applications. However, the cyclic responses of this TPMS structure used in bearings have never been completely investigated. This study is the world's first to investigate the effects of normal pressure on the cyclic responses of novel 3D-printed TPMS bridge bearings. A numerical TPMS unit cell model considering the effects of normal pressure on cyclic responses of a novel TPMS bridge bearing is developed and validated with experimental data. The numerical results reveal new insights related to the nonlinear effects of normal pressure on the cyclic behaviours of 3D-printed TPMS bearings. Higher normal pressures result in a higher degree of nonlinearity in the dynamic cyclic responses of the 3D-printed TPMS bearings.

**Keywords:** crashworthiness; specific energy absorption; triply periodic minimal surface (TPMS); a novel 3D-printed TPMS bridge bearing





## 1. Introduction

Over 70 years, bearings have been widely used in bridge system as they are able to transfer/accommodate the loads/displacements in vertical and horizontal directions between the superstructure (girders) and the substructure (columns) [1,2]. In bridge bearing applications, they can also experience rotational displacements [3,4]. It is important to design these typical bridge bearings with sufficient vertical stiffness and lateral flexibility, in order to support the weight of the superstructure of a bridge and facilitate horizontal/rotational deformations induced in girders [1,5–7]. Thus, the base isolation of bearings is an alternative method to reduce the seismic demand in bridge systems [8,9]. In a critical review [10], bridge bearing failure can potentially lead to accelerated bridge deterioration or result in bridge damage. As such, this damage is prevented in bridge bearings subjected to any expected loading condition.

The practical use of a bearing type is based on several factors, including geometry, deflection, predicted loading, maintenance, displacement and rotation demands, existing clearance, strategies, designer preference, and availability, as well as cost [11]. Accordingly, standard bridge bearings can be classified into two groups: either elastomeric bearings or high-load multirotational (HLMR) bearings (spherical, pot, and disc) [12]. An elastomeric bearing is a combination of elastomeric pads (rubber layers) without/with any reinforcement (steel shims or fibre fabric), called plain and reinforced elastomeric bearings,

respectively. The reinforcement has a high tensile capacity, in order to increase the vertical stiffness of the bearing by limiting bulging of the rubber.

Nevertheless, common bridge bearings are still likely to have a bulging behaviour under compression and to experience the stress concentrations at the edges of interactions between reinforcement and rubber layers. In a case of a fully bonded bridge bearing, if the cyclic demands of bearing exceed a certain capacity limit, the bearing failure will be initiated, resulting in the internal rupture of the bearing, which begins to propagate towards until the entire delamination [13]. These two phenomena might cause the bridge to fail immediately if their design is insufficient under compression and cyclic condition. Furthermore, the stress concentrations in the bridge bearings with steel reinforcement can potentially lead to the failure of delamination at the interaction layers between steel shims and rubber. In fact, the installation approaches of bridge bearings in bridge systems around the world can be found as fully bonded, fully unbonded, and single-side bonded. In this paper, only the cyclic response of a fully bonded bridge bearing is considered. According to a review in [13], there are two major reasons to investigate the proposed model for a fully bonded installation: One is because under extreme cyclic loading, the bearings with a fully bonded installation, among others, are most useful to restrain the deformations of the superstructure due to the fully high capacity of their shear stiffness. Another reason is that these fully bonded connections of bridge bearings are very compatible with any steel bridges, which are widely used due to highly economic costs, facilitating the mechanical connections between the bearings and bridges.

Further drawbacks of using typical bridge bearings are that they are hard to be recycled and have high labour costs, as well as heavy weight with steel and rubber materials. It is clear that these bearings have a low performance-to-weight ratio when subjected to any loading conditions. At this point, in terms of better designs for bridge bearing applications, the development of 3D-printed TPMS bridge bearings under any static and dynamic loading has been inspired by eliminating the aforementioned problems. Based on our previous works [1,14–18], the use of 3D-printed TPMS structures, which have a relatively higher performance-to-weight ratio, has been considered for bridge bearing applications, resulting in improving their behaviours under any anticipated loading, compared to traditional bearings.

Meanwhile, the help of additive manufacturing technology (3D printing) offers the opportunity to realise TPMS structures with complex designs and features. These structures are minimal surfaces that are periodic in three-coordinate directions with zero mean curvatures, free of self-intersections. Furthermore, 3D-printed TPMS structures have better properties compared to common structures as continuous curves, resulting in previous layers supporting the following layers without support during the additive manufacturing process [19–21]. Unlike other lattice structures, they are required for their support or limited by their angle [22].

Regarding the benefits of using 3D-printed TPMS structures for bridge bearing applications, Pasakorn and Sakdirat [15] have studied the determination of mechanical properties and energy-absorption capability of a 3D-printed triply periodic minimal surface (TPMS) sandwich lattice bridge bearing model subject to combined compression–shear loading. It is found that the proposed model can better behave under the combined compression–shear loading than a common bridge bearing model due to relatively higher shear capacity and specific energy absorption. Additionally, its failure modes observed via the stress–strain curve of this novel TPMS bridge bearing model are identified as the hysteretic failure behaviour under the same loading condition.

According to bridge bearings' operative performance in bridge systems, the increase in normal pressure on the bearings under cyclic loading conditions is one of the main factors affecting the response of the bridge. This circumstance causes a reduction in shear modulus or shear stiffness when the normal pressure increases [23,24]. In addition to the performance of bearings, the force–displacement hysteresis loops of bearings become wide,

exhibiting stable and powerful energy dissipation capacity, when the bearings start to slide on the concrete substructure with an initial friction coefficient of 0.25–0.5 [25].

As such, the compressive response of the TPMS bridge bearing is required to be extensively investigated to observe the TPMS structure design to prevent its buckling or yield failures in compression due to its porous geometry, but the cyclic response of the bearings to the bridge also needs to be considered. This means that they can continue to accommodate any horizontal displacements under cyclic loading or thermal movement conditions, if the proposed bearing does not buckle in compression first. It is necessary to deeply understand and predict the shear behaviour under cyclic loading of this novel bridge bearing with the help of the TPMS structure possessing a relatively higher performance-to-weight ratio, in order to improve its mechanical properties.

To the best of our knowledge, following a critical review of the literature available in bridge bearing applications, no numerical models have been proposed to describe the cyclic response of a novel 3D-printed TPMS bridge bearing, especially when the bearing is affected by the increase in normal pressures. The shear response of a novel 3D-printed TPMS bridge bearing is likely to be fairly sensitive to the variation in axial load. This variation in axial load is mostly induced by the normal pressure excitation during cyclic loading.

This paper describes a series of simulations of a novel 3D-printed TPMS bridge bearing used for any small- to medium-span highway bridges around the world. A proposed model is designed and developed based on the experimental results [26] to correctly predict its cyclic behaviour. The theory and numerical model design, including materials and methods, will be detailed in the following section, followed by the comparative results and discussion of the effects of normal pressures on the proposed model under cyclic loadings. These insights will be useful for further developing TPMS bridge bearings under seismic loading.

## 2. Theoretical Background for the Cyclic Response of a Novel TPMS Bridge Bearing (NTBB)

*Fully Bonded Novel TPMS Bridge Bearing*

Before investigating the effects of normal pressures on the cyclic responses of a novel TPMS bridge bearing, it is necessary to identify all different force components acting on the NTBB imposed to cyclic loading. Generally, the possible force components consist of the compression and shear forces, pure bending moment, tensile force, and vertical force-induced bending moment. As can be seen in Figure 1, during cyclic loading conditions, the compression and shear force are observed to act on the NTBB, which also supports the weight of a bridge superstructure at the same time. In most cases of the pure bending moment, it occurs when the top surface of the superstructure rotates around the transverse axis because of the considerable flexural deformation. In addition to this component, it is unlikely to appear a pure moment phenomenon for such stiff structures as short piers and abutments. However, they can face this phenomenon when their substructure foundations fail. The tensile force of the NTBB will occur in a case of continuous superstructure with nonuniform settlement between adjacent substructures. In terms of the vertical force-induced bending moment, it typically increases from the P-Delta effect due to the combination of a dead load and horizontal displacements.

Figure 1 demonstrates the force diagram for the novel bonded TPMS bridge bearing, which is useful to comprehend its failure pattern. As seen from Figure 1, in any case of a flexural deformation-dominated substructure, the fully bonded TPMS bridge bearing subjected to combined compression–shear loading condition would be also imposed to the pure bending moment that results from the rotation at the upper surface of the substructure. Therefore, all these three load components would create bending moments around the neutral axis on the bearing. The superimposition is used for the bending moments resulting from various sources; the biggest moments would appear at the upper and lower sides of the bearing, at which the connecting set bolts are located. Thus, they are prone to high and complex local stresses which is the major cause of bolt fracture. Note that the bolt fracture damages are not considered in this study. This is because the rupture of the TPMS

structures used for bridge bearing applications is likely to be observed first before that of bolts, due to their porosity. The design of a proposed 3D-printed TPMS bridge bearing representing the novel TPMS bridge bearing will be described in the following section.

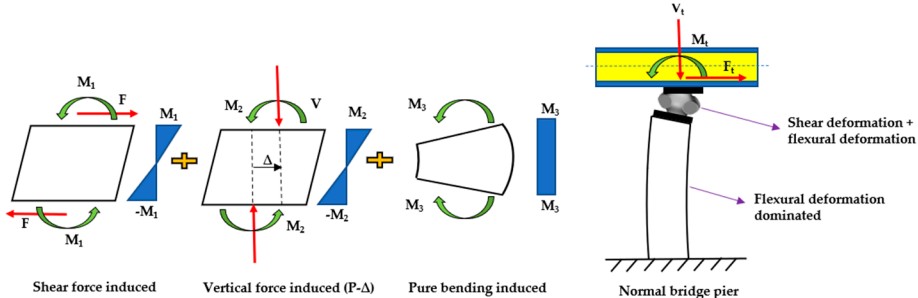

**Figure 1.** Force diagram for a fully bonded novel TPMS bridge bearing with flexural deformation-dominated bridge substructure.

## 3. A Proposed 3D-Printed TPMS Bridge Bearing (TPMSB)

### 3.1. Proposed TPMS Bridge Bearing Specimens

From our previous work [15], three schwarz primitive (SP) unit cell specimens (50 mm × 50 mm × 50 mm) are fabricated using a stereolithography (SLA)-based 3D printer, as shown in Figure 2, and then they are tested subjected to uniaxial compression. Meanwhile, in this paper, the previously proposed model using an SP structure will be further designed, validated, and investigated under cyclic loading with normal pressure variations in the following sections.

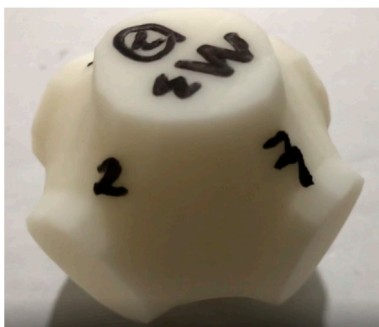

**Figure 2.** The SP unit cell bearing specimen manufactured by a 3D printer.

### 3.2. A Proposed TPMS Bridge Bearing Model

Figure 3 shows the proposed 3D TPMS bridge bearing lattice model using several schwarz primitive (SP) cellular structures without thickness. Basically, the proposed TPMS bridge bearing model can be represented as a combination of several SP unit cells, which can provide the same behaviours under any loading conditions without considering the geometry sizes. Schwarz surface is one of the most used triply periodic minimal surfaces (TPMSs) for multiple applications, which provides a relatively high performance-to-weight ratio when subjected to compression. Basically, these TPMS structures can be created with any approaches. For example, in this paper, we use the first approach, which generates a network struct throughout an SP structure for bridge bearing applications. Furthermore, it is a well-known skeletal Schwarz primitive for the proposed bridge bearing model without thickness, based on one of the subdomains divided by the surface as a solid. Another approach is to generate a termed sheet-based or double Schwarz primitive for modelling an SP bridge bearing with thickness, by plotting two minimal surfaces with two variant level sets of the constant j together, resulting in an offset from a hypothetical surface at the average of the two level-set surfaces. The second method to nearly create an SP structure model is dependent on the following equation:

$$\cos\left(\frac{2\pi x}{t}\right) + \cos\left(\frac{2\pi y}{t}\right) + \cos\left(\frac{2\pi z}{t}\right) - j = G(x, y, z) \tag{1}$$

where *t* defines the unit cell size of a lattice (Figure 3) and *j* refers to the volume fraction of the areas that is divided by the surface [27]. The volume fraction of a lattice describes the relative density of its elements. It can also be employed to modify the thickness of the model and to compare models with the same volume. It is interesting to mention that in this paper, we only utilise the first approach for creating a skeletal SP bridge bearing model. Figure 4 demonstrates a Schwarz primitive unit cell CAD model (a 50 mm × 50 mm × 50 mm volume) used for bridge bearing applications with a combination of the several ones in this paper. The 3D-printed bearing unit model is designed, conforming to the bearing standard used in Thailand [28].

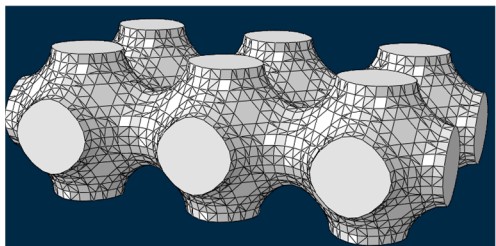

**Figure 3.** Schwarz primitive lattice CAD model with the help of the SP unit cell combination.

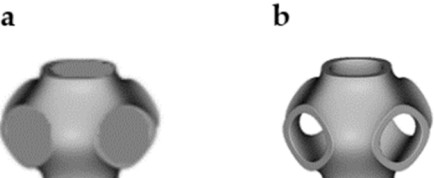

**Figure 4.** Schwarz primitive unit cell CAD model without (**a**) and with (**b**) thickness, respectively.

It is important to mention that the proposed SP bridge bearing model created is considered using the first approach, because of a relatively higher bearing loading-to-weight ratio under compression among the other TPMS unit cells, in order to make sure this bearing model can support the weight of a bridge superstructure before experiencing horizontal displacements under shear.

### 3.3. Material Model

According to our previous work [15], the proposed photosensitive resin (UV resin) material possessing rubber-like properties is chosen to be manufactured for a SP unit cell bridge bearing for compression testing, and the validation is also conducted for both static compression and shear analysis only. To perform simulations of this proposed model under cyclic loading with normal pressure variations, the model material is calibrated again with help of experimental results [26]. The model validation will be presented in the following section. Figure 5 shows the stress–strain curve of the proposed material used in the simulations of the model SP unit cell bridge bearing under cyclic loading. It is assumed that the material model has homogenous properties. It is to be noted that these stress–strain relationships from the curve below are calibrated in Abaqus [29] as a bilinear curve with an approximate yield point to obtain correct data when simulating due to its elastic–plastic behaviour.

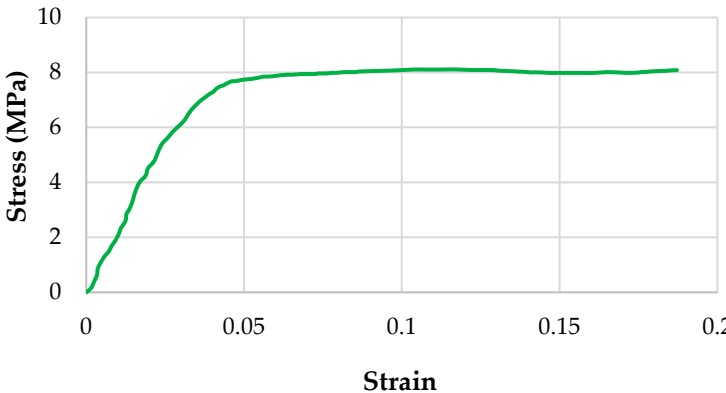

**Figure 5.** Proposed material properties from combined compression–shear stress–strain relationships used in all simulations.

*3.4. Methods*

3.4.1. Simulation Program

The whole simulation program is summarised in Table 1. The loading displacement is represented by the equivalent shear strain (ESS), which is the value of the applied displacement on the centre location of the proposed model's upper surface. The loading procedure in the simulations is monotonic or dynamic cyclic with the displacement in the unit of ESS. The normal pressure on the proposed model varies during all the simulations to examine sensitivity in the model responses to this parameter. The normal pressure on the model ranges from 0 to 3.5 MPa, just within the design bridge bearing capacity. It is important to note that the proposed model does not fail in buckling for the aforementioned pressure range, hence Euler's buckling effect can be negligible.

**Table 1.** Simulation programs.

| Simulation Series | Program | Normal Pressure (MPa) | ESS (%) |
|---|---|---|---|
| 1-1 | Monotonic | 0.0 | 50 |
| 1-2 | Monotonic | 0.5 | 50 |
| 1-3 | Monotonic | 1.0 | 50 |
| 1-4 | Monotonic | 1.5 | 50 |
| 1-5 | Monotonic | 2.0 | 50 |
| 1-6 | Monotonic | 2.5 | 50 |
| 1-7 | Monotonic | 3.0 | 50 |
| 1-8 | Monotonic | 3.5 | 50 |
| 2-1 | Cyclic | 0.0 | 50 |
| 2-2 | Cyclic | 0.5 | 50 |
| 2-3 | Cyclic | 1.0 | 50 |
| 2-4 | Cyclic | 1.5 | 50 |
| 2-5 | Cyclic | 2.0 | 50 |
| 2-6 | Cyclic | 2.5 | 50 |
| 2-7 | Cyclic | 3.0 | 50 |
| 2-8 | Cyclic | 3.5 | 50 |

Several dynamic shear tests were conducted on common/reinforced bridge bearings [30] using a horizontal displacement of increasing amplitude from 10 mm to 45 mm, with a frequency of 0.87 Hz. Each amplitude level includes three fully reversed cycles. However, to obtain the cyclic behaviour of the SP unit cell bridge bearing model in this paper, the horizontal displacement is reduced to 30 mm due to its smaller geometry compared to that of the bearing specimens. This is because the proposed SP unit cell bridge bearing has a length of 50 mm and the target shear stain is designed for 50%. Furthermore, the design vertical pressure of the model is 1 MPa.

Figure 6 illustrates the applied displacement time history used for all the simulations of the model SP unit cell bridge bearing. It is to be noted that in this paper, the shear behaviour of the novel TPMS bridge bearing without the flexural deformation-dominated bridge substructure is only investigated, and the P-Δ effect on its cyclic response is also out of the scope. This means that the combined compression and shear forces are only considered for the cyclic loading condition. It is also assumed that the moments induced by the forces in this condition are also negligible due to the model being fixed in all three rotational directions.

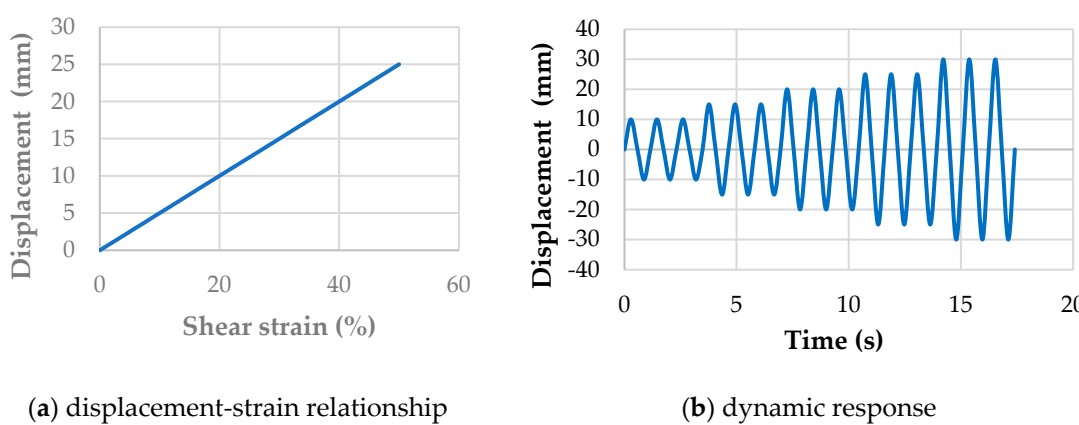

(**a**) displacement-strain relationship          (**b**) dynamic response

**Figure 6.** (**a**) applied horizontal displacement against shear strain (**b**) applied horizontal displacement against time history used for all the simulations of the model SP unit cell bridge bearing under both monotonic and cyclic loading, respectively.

3.4.2. Model Validation

To evaluate the proposed numerical model of an SP unit cell bridge bearing considering the effects of normal pressures on its cyclic responses, a comparison is performed between the numerical simulations and experimental results previously referred to. The proposed model is investigated according to the cyclic loading procedure used during testing of the recycled rubber fibre-reinforced bearings (RR-FRBs) with 3D dimensions of 70 mm × 70 mm × 63 mm. The equivalent horizontal stiffness, $K_{eq}$, of both the model and the RR-FRB at a 25 mm displacement is calculated using Equation (2). The model validation is performed as presented in Figure 7. Generally, the results shown in Figure 7 indicate a good agreement between the proposed model and experimental results, with less than 4% and 8% for equivalent stiffness, $K_{eq}$, and dissipated energy, $D_{en}$, respectively, especially for their equivalent horizontal stiffness at a 25% shear strain in Table 2.

Nevertheless, the proposed model is not able to completely model the hyperelastic effect that appears in the dissipated energy during cyclic loading when the loading direction changes. This is because the proposed model material is considered as an elastic–plastic behaviour. It is obvious that the load–displacement curve of the proposed model is wider than that of a common bridge bearing tested due to a relatively higher yield point.

In terms of dissipated energy, Table 2 also lists the comparison of dissipated energy between the experimental and numerical results for cyclic loading. The dissipated energy of a bearing is basically determined based on the area of the hysteresis loops obtained from the experiments or numerical analyses. As seen in Table 2, the difference is small, just lower than 8%. The proposed model can describe the energy-dissipative behaviour of a novel 3D-printed TPMS bridge bearing under cyclic loading with reasonable accuracy.

Additionally, several major responses of a novel TPMS bridge bearing can be captured by the proposed model, such as dynamic shear behaviour under cyclic loading. Thus, the cyclic response of the novel bridge bearing using a combination of SP unit cells can be reasonably predicted by the proposed model in the numerical analysis.

$$K_{eq} = \frac{F_2 - F_1}{d_2 - d_1} \tag{2}$$

where $d_1$, $d_2$ are applied horizontal displacements of the proposed model at point 1 and point 2, respectively (mm), and $F_1$, $F_2$ denote the corresponding shear force (N) at $d_1$ and $d_2$, respectively.

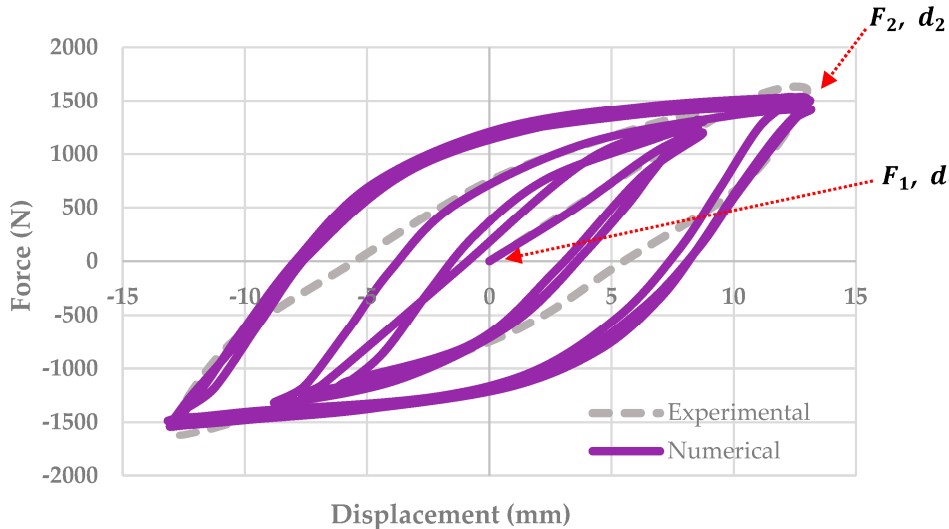

**Figure 7.** Comparison of shear behaviours of the two similar bridge bearings under cyclic loading at a 25% shear strain between numerical and experimental data.

**Table 2.** Showing a comparison of the equivalent horizontal stiffness ($K_{eq}$) and the dissipated energy ($D_{en}$) of the proposed model at a 25% shear stain between the experimental and numerical results.

|  | Equivalent Stiffness, $K_{eq}$ (N/mm) | Dissipated Energy, $D_{en}$ (N·mm) |
| --- | --- | --- |
| Experimental | 123.08 | 61,175 |
| Numerical | 118.17 | 56,346 |
| Difference ratio (%) | 3.99 | 7.89 |

## 4. Results

### 4.1. Observed Performance

With the normal pressure on the SP unit cell bridge bearing model maintained on a specified value, the gradually increasing horizontal displacement is imposed on the proposed model. As seen in Figures 8 and 9, for the proposed model's performance subjected to designed pure compression (1 MPa) before shear resistance, the model does not yield in compression and also continues to experience shear loading. However, it starts to yield in compression when the normal pressure is higher than 1.5 MPa, as seen Figure 8. In terms of monotonic shear loading condition, the proposed model exhibits smaller shear deformation at small shear demand (etc. ESS = 25%), with some of the upper and lower parts yield under combined compression–shear loading. When the ESS reaches 50% (target shear), the proposed model becomes relatively wider as a block structure (crashworthiness behaviour) compared to that at lower shear strain rate, in order to experience a higher shear strain rate, as indicted in Figure 8, especially for the increasing normal pressure up to 3.5 MPa, seen in Figure 9.

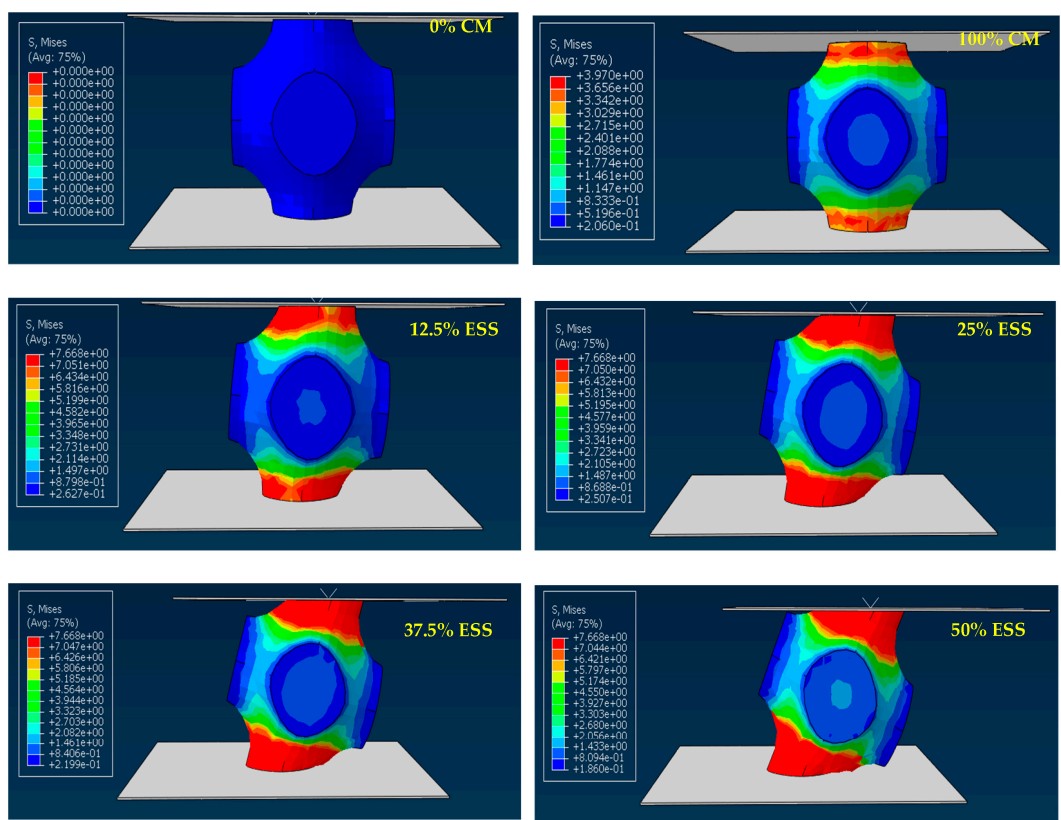

**Figure 8.** Schematic shear behaviours of the proposed model for monotonic loading procedure at different shear strains with a designed normal pressure of 1 MPa.

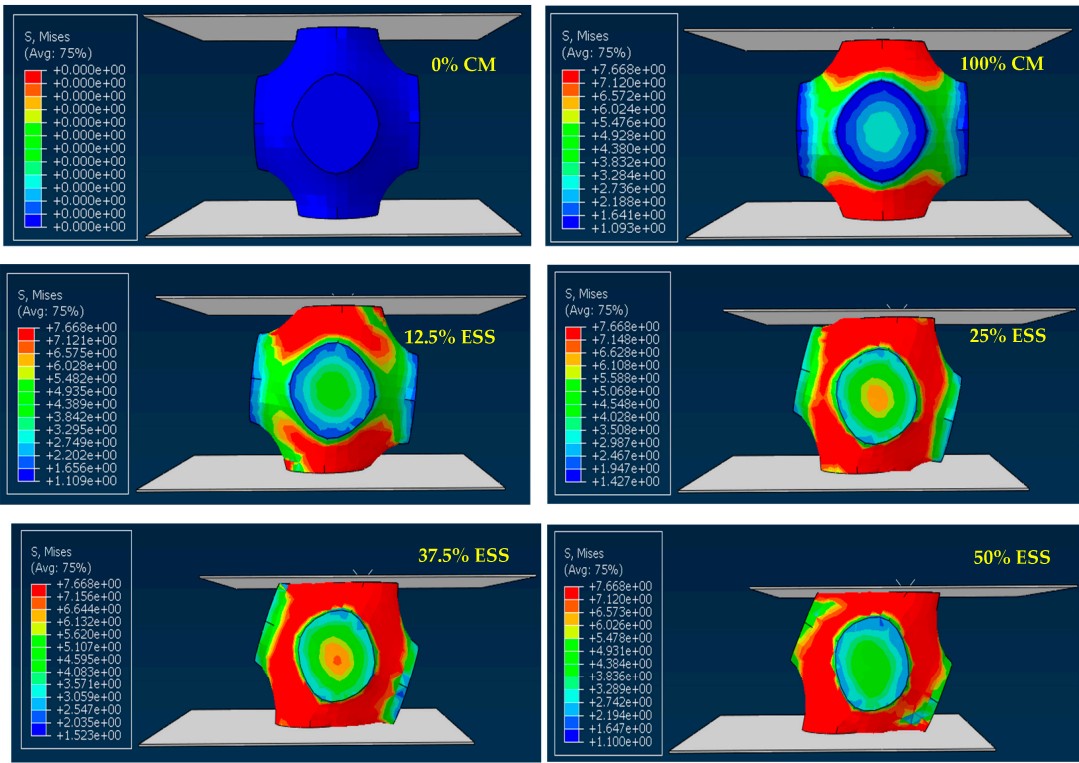

**Figure 9.** Schematic shear behaviours of the proposed model for monotonic loading procedure at different shear strains with a designed normal pressure of 3.5 MPa.

Considering the proposed model sliding, it is clear that the upper and lower surfaces of the model are in full contact with rigid plates under any loading conditions, due to their fully bonded condition (Figures 8 and 9). With the applied displacement increases, the shear deformation appeared in the proposed model increases slightly. It is to be noted that the frictional sliding response of the proposed model or buckling instability is not observed in the model during the simulations; therefore, the proposed mode is proved to conduct enough, despite being imposed to target-rate cyclic demands.

Furthermore, the stress distribution of the proposed model under monotonic loading with normal pressure variations between 1 MPa and 3.5 MPa, seen in Figures 8 and 9, is identical at the initial state (12.5% ESS). After this, the proposed model considerably resists the relatively higher stresses induced in the whole part, resulting in the form of a structure block for accommodating higher horizontal displacements. It is important to note that the stress distribution of the proposed model subject to cyclic loading is the same pattern.

*4.2. Force–Displacement Response*

Force–displacement response curves for the proposed model subjected to monotonic procedure with variations in normal pressures are plotted in Figure 10 to present their static shear behaviour at low-to-target shear strains (50% ESS). Three ellipse markers are drawn individually to show the states where the deformation and stress distribution are initiated in elastic, plastic, and higher strain-hardening regions being wider in the horizontal direction, respectively. These markers are employed to categorise the three phases of the proposed model responses. According to Figure 10, the elastic behaviour of the proposed model is relatively insensitive to normal pressure at Phase 1, where only shear deformation in the top and bottom column parts occurs. With the increase in normal pressure, this leads to a reduced secant horizontal stiffness at Phase 2, but an increased one at Phase 3 due to its crashworthiness behaviour.

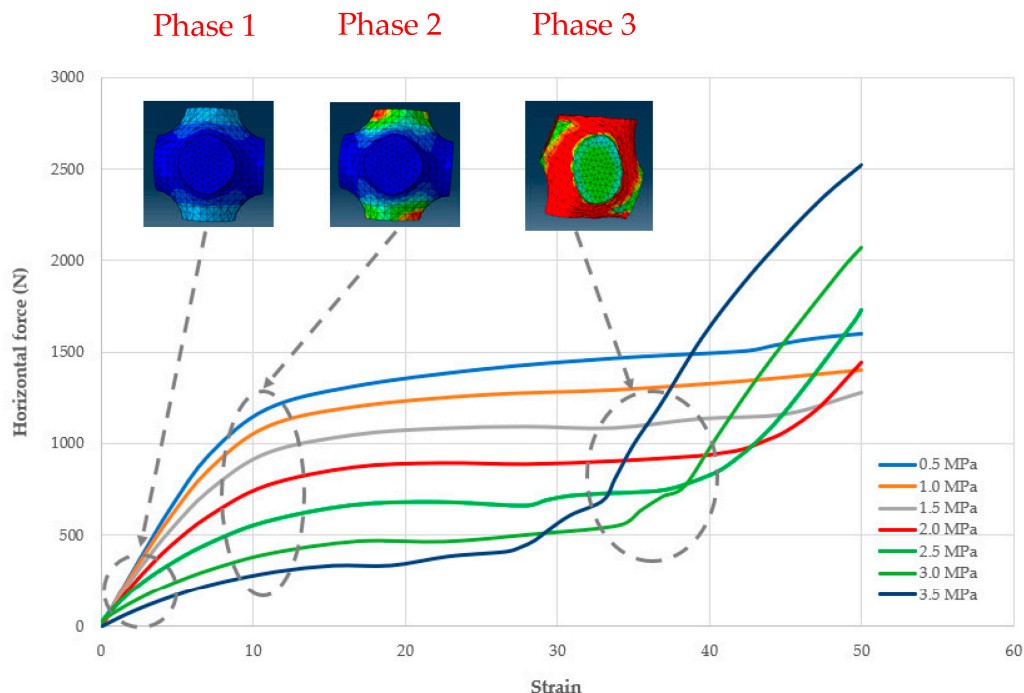

**Figure 10.** Force–displacement curves of the proposed model for monotonic loading procedure.

The predicted force–displacement hysteresis curves of the proposed model are plotted in Figure 11, in which Figure 11a,b,c,d,e,f,g and h are corresponding to 0, 0.5, 1, 1.5, 2, 2.5, 3, and 3.5 MPa, respectively. When the amplitudes of applied shear strains are small, the proposed model deforms absolutely in shear and exhibits narrow hysteresis loops,

accounting for the nonlinear effects in its proposed material and complex porous structure. Nevertheless, the response of the proposed model at small shear strains is potentially considered as linear elastic, which is satisfied for this simulation modelling. Since the shear strains increase, the hysteresis loops become relatively wide and stable, showing decent energy dissipation in the proposed model under cyclic loadings. The increase in normal pressures causes development of a stiffening effect due to the change in applied loading displacement. Moreover, this effect is obvious when compared to the hysteresis loop of the proposed model under pure cyclic loading, as found in the Section 5. Despite the stiffening effect, the unloading branches of the hysteresis loops are typically parallel to the initial loading or reloading branches, demonstrating a decent match in shear stiffness.

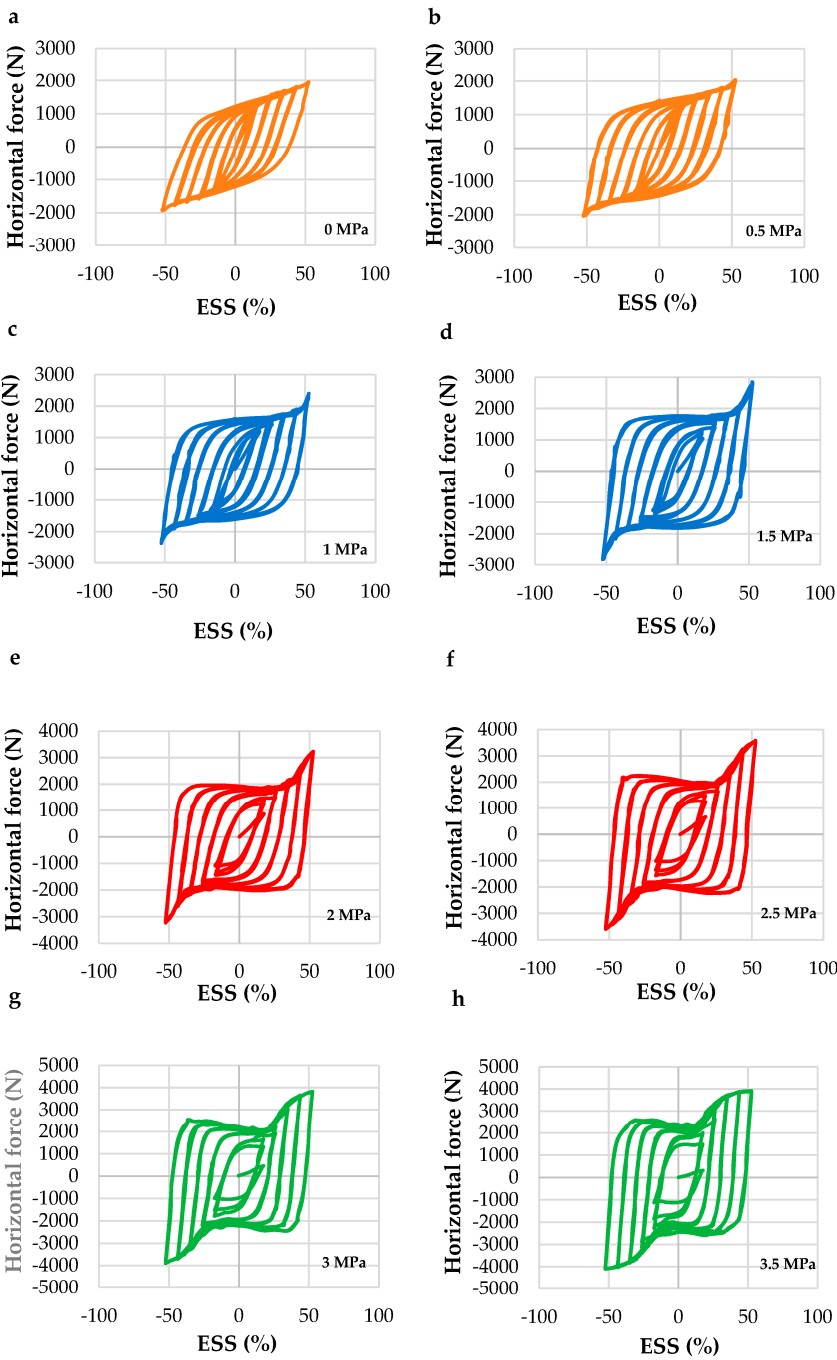

**Figure 11.** Force–displacement hysteresis curves of the proposed model subjected to cyclic loading without/with normal pressure variations of 0.5 up to 3.5 MPa.

*4.3. Bearing Shear Stiffness and Shear Deformation*

Typical bridge bearings are characterised by facilitating girder movements from thermal expansion without any observed distress, particularly for noncyclic conditions. These bearings are expected to exhibit elastic behaviour during bridge operations, despite the material nonlinearity in rubber. Another nonlinearity of TPMS structures for bridge bearing applications in this paper that is considered is the complex porous geometry of the proposed model. This is likely to affect its characteristics, resulting in a change in TPMS bearing responses during bridge operations. Therefore, it is required to investigate the unexpected responses of these TPMS bearings having nonlinear properties. The bearing stiffness is one of the most important factors, which expresses the mechanical behaviour of a SP unit cell bridge bearing model for bridge applications. Basically, the stiffness of the proposed model is based on terms of shear modulus. The shear modulus can be calculated using the following equation related to many different standards, where most of them specify the approaches evaluating the shear modulus from the force–displacement curves (hysteresis loops). The effective (equivalent) modulus of elasticity is defined to nearly represent the whole shear modulus of the proposed model, shown in Figure 12. The effective shear modulus, G (MPa), from zero shear strain to a point, which is the boundary between Phase 1 and Phase 3, is obtained based on the following equation [23].

$$G = \frac{\Delta F H}{\Delta d A} \qquad (3)$$

where $\Delta F$ *and* $\Delta d$ are the horizontal force and displacement from zero shear strain to a point, respectively, and *H* defines the total thickness of the proposed model (mm); *A* is its plan area (mm$^2$).

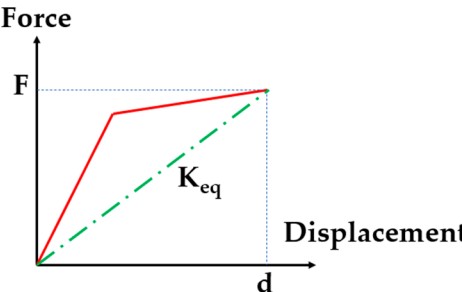

**Figure 12.** Definition of effective stiffness. Red line is the bi-linear relationship between force and displacement, while the green dash line is the equivalent stiffness.

Table 3 summarises the effective shear moduli of the proposed model under monotonic and cyclic loadings with normal pressure variations. These shear moduli are calculated using the above-mentioned equation, in order to give context for investigating the performance of the proposed model with regard to both the conditions. For monotonic analyses, Figure 13 indicates that the effective shear moduli of the proposed model, $G_{em}$, are observed to reduce with increasing shear strains at the initial response. The reduction in shear modulus or shear stiffness at large strains is mostly dominated by two key factors: the proposed material model and nonlinearity geometry. These shear moduli are found to be more sensitive to the variation of normal pressure stress when the normal pressure reaches 2 MPa at high strains.

The calculated effective shear moduli of the proposed model for cyclic analyses at different normal pressures are summarised in Table 3. According to Figure 13, the effective shear modulus, $G_{ec}$ will normally increase with the increase in normal pressure for cyclic analysis. Unlike the other one, the effective shear modulus, $G_{em}$ will basically decrease with the initial increase in normal pressure. When the normal pressure is higher than 1.5 MPa, the effective shear modulus increases because of the change in structure shape like a typical bearing's block. It is important to note that the significant difference in the shear moduli of

the proposed model between monotonic and cyclic conditions is fairly acceptable due to the effect of static and dynamic loadings on the behaviours of the proposed model [31,32]. The dynamic shear force obtained from cyclic analysis at 50% ESS is 1.6 times higher than the static shear force resulting from monotonic analysis at the same ESS, seen in Figures 10 and 11. Another reason is that the proposed model subjected to cyclic loading will continue to resist higher shear strains induced the next cycle after the previous cycle.

**Table 3.** Calculated shear moduli of the proposed model under monotonic and cyclic loading at the target shear stain.

| Normal Pressure Variation (MPa) | $G_{em}$ (MPa) | $G_{ec}$ (MPa) | ESS (%) |
|---|---|---|---|
| 0.0 | 1.80 | 1.88 | 50 |
| 0.5 | 1.63 | 1.96 | 50 |
| 1.0 | 1.43 | 2.28 | 50 |
| 1.5 | 1.31 | 2.73 | 50 |
| 2.0 | 1.47 | 3.13 | 50 |
| 2.5 | 1.76 | 3.47 | 50 |
| 3.0 | 2.11 | 3.70 | 50 |
| 3.5 | 2.57 | 3.81 | 50 |

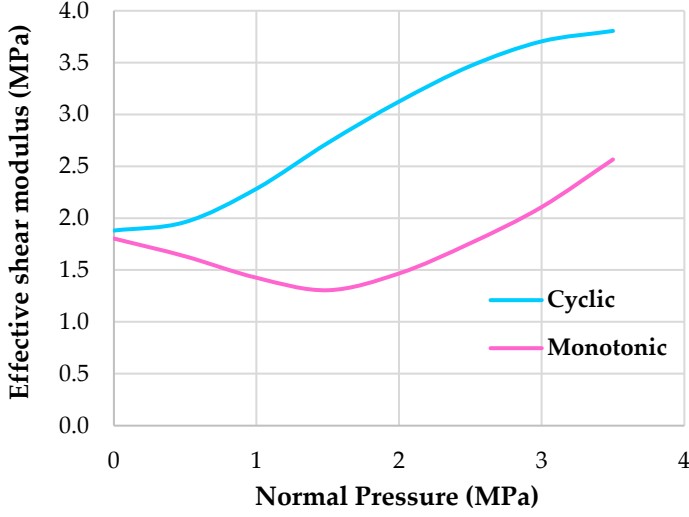

**Figure 13.** Summary of the calculated effective shear moduli of the proposed model at different normal pressures for both monotonic and cyclic analysis.

## 5. Discussion

Figure 14 demonstrates comparative force–displacement hysteresis curves of the proposed model used for bridge bearing applications under cyclic loading conditions with normal pressure variations, at the target shear strain (50% ESS). The shear moduli are likely to vary in each simulation, as they are observed and determined in each loading cycle. In addition, only the upper bound value for the shear modulus is evaluated in Table 3 above. The comparison of different shear behaviours in cyclic simulations is identical to those in monotonic ones without unloading and reloading.

As seen in Figure 14, the effective shear moduli of the proposed model reduce with the increasing number of cycles. The upper bounds of shear moduli with the variation of normal pressure represent the values obtained from the last loading cycles. The reduction in shear stiffness with the loading cycles is mostly due to the material model owning an elastic–plastic-like material property. Unlike a rubber-like material, it possesses a softening material property and can cause a nonlinear phenomenon during high loading and unloading paths, called the Mullins effect. The shear strain values of the model are also observed to increase with the increase in cycle numbers, because of strain hardening.

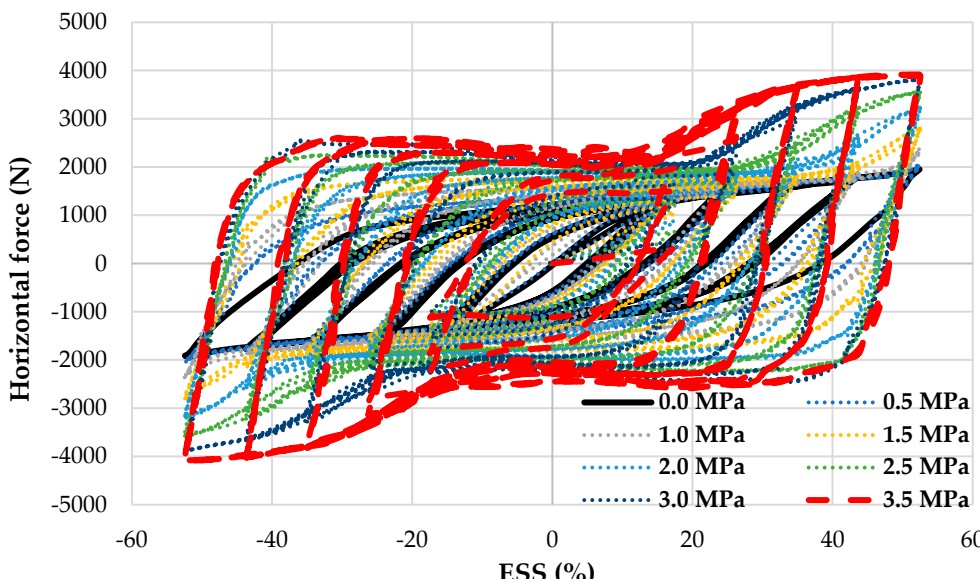

**Figure 14.** Illustrating force–displacement hysteresis curves of the proposed model for cyclic analyses without/with normal pressure variations.

Surprisingly, the initial shear modulus (secant one) of the first circle for each simulation is found to significantly decrease with the increasing normal pressure, compared to that under pure cyclic loading (without normal pressure), seen along the black line in Figure 14). This might be due to the proposed model, which experiences its yield first under relatively higher normal pressure compression before accommodating any horizontal displacements for cyclic conditions. After the first circle, the secant shear moduli increase due to the proposed model beginning to yield under compression and being wider in horizontal direction. It is assumed that the buckling or P-Delta deflection has no effect on the proposed model's responses during cyclic analysis due to its fully bonded connections. In fact, the proposed model is prevented from buckling failure due to its low slenderness ratio. The new insight is essential to establish a comprehensive digital twin for bridge monitoring and maintenance [33].

## 6. Conclusions

A series of simulations are conducted to investigate the cyclic response of a novel 3D-printed TPMS bridge bearing with the help of a combination of TPMS unit cells, which can be considered as the proposed TPMS unit cell model, due to the novel bearing's structure owning a periodic cellular pattern. The 3D-printed bearing is designed and fabricated conforming to the bearing standard used in Thailand. The simulations of the proposed model include various combinations of normal pressure and shear loading analyses (monotonic and fully reversed dynamic cyclic). Applied horizontal displacements and vertical loads are used in all the simulations for both conditions at a target level. A maximum target shear strain of 50% is reached for the monotonic and cyclic loading conditions. Conclusions are listed from the present works as follows:

(1)　The proposed model is prone to mimic the cyclic behaviour of a typical bridge bearing for bridge bearing applications, as well as nearly offering an identical dissipated energy for bridge bearing applications, as seen in Figure 7 and Table 2, respectively. The model can act exactly as a cellular rubber block structure of a bridge bearing, transferring/facilitating horizontal forces/displacements between the superstructure and the substructure while supporting the weight of the superstructure.

(2)　The difference in the von Mises stress distribution of the proposed model between 1 and 3.5 MPa is observed and found to have an increasing trend with the increase in normal pressure (Figures 8 and 9). The distribution pattern is the same at the initial

state (12.5% ESS), and higher stress distributions are found over the whole model when the model's applied horizontal displacement reaches the target shear strain (50% ESS).

(3)     In terms of the TPMS bearing's characteristics, for dynamic cyclic analysis, the effective shear moduli of the proposed model are observed to have an increasing trend with the increase in normal pressures. Unlike monotonic analysis, its effective shear moduli are found to decrease in the initial phase until the normal pressure is more than 1.5 MPa. This is because the nonlinearity of the complex TPMS structure mainly changes the structure shape to be wider horizontally (a column into a block) in order to resist higher shear strains. The cause of the structure change into the rubber block-like structure (the well-known crashworthiness behaviour) is that the model initially experiences a higher yield stress while experiencing shear.

(4)     The effective shear stiffness of the proposed model under the cyclic loading conditions trends downward with the large number of cycles, as indicated in Figures 11 and 14.

(5)     The better performance of the proposed model is also found to offer strain hardening with the increasing number of cycles due to the material model being considered as an elastic–plastic behaviour, which differs from the Mullins phenomenon that occurs in common elastomeric bridge bearings. This leads to the increase in shear strength of the proposed model after unloading or reloading during repetitive cycles.

(6)     The effective dynamic shear moduli for cyclic loadings are 1.6 times higher than those for monotonic loadings, because the proposed model behaves immediately under dynamic cyclic loading conditions. For example, the dynamic shear force of 4 kN at a 3.5 MPa normal pressure is more than that of the static force of 2.5 kN (Figures 10 and 14, respectively), resulting from the dynamic shear modulus, which is higher than the static one for monotonic loading conditions. Furthermore, the load patterns between both conditions make different initial responses with the increase in normal pressure.

**Author Contributions:** Conceptualisation, P.S.; methodology, P.S.; software, P.S.; validation, P.S.; formal analysis, P.S.; investigation, P.S.; resources, P.S; writing—original draft preparation, P.S.; writing—review and editing, S.K.; visualisation, P.S.; supervision, S.K.; project administration, S.K.; funding acquisition, S.K. All authors have read and agreed to the published version of the manuscript.

**Funding:** This research was funded by the European Commission, grant number: H2020-MSCA-RISE No. 691135 and H2020 Shift2Rail Project No. 730849.

**Data Availability Statement:** The data that support the findings of this study are available from the corresponding author upon reasonable request.

**Acknowledgments:** The first author wishes to thank the Royal Thai Government for his Scholarship at the University of Birmingham. The last author wishes to gratefully acknowledge the Japan Society for Promotion of Science (JSPS) for his JSPS Invitation Research Fellowship (long term), Grant No. L15701, at the Track Dynamics Laboratory, Railway Technical Research Institute and at Concrete Laboratory, the University of Tokyo, Tokyo, Japan. The JSPS financially supported this work as part of the research project entitled "Smart and reliable railway infra-structure". Special thanks are given to the European Commission for H2020-MSCA-RISE Project No. 691135 "RISEN: Rail Infrastructure Systems Engineering Network" (www.risen2rail.eu), accessed on 16 August 2021. Partial support from H2020 Shift2Rail Project No. 730849 (S-Code) is acknowledged. In addition, the sponsorships and assistance from LORAM, Network Rail, RSSB (Rail Safety and Standard Board, UK) are highly appreciated.

**Conflicts of Interest:** The authors declare no conflict of interest.

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
