# Peer review of "Investigations into Nonlinear Effects of Normal Pressures on Dynamic Cyclic Responses of Novel 3D-Printed TPMS Bridge Bearings"

_vibration, doi:10.3390/vibration6010006_

Round 1

Reviewer 1 Report

In this paper a numerical triply periodic minimal surface (TPMS) unit cell model considering the effects of normal pressure on cyclic responses of a novel TPMS bridge bearing has been developed. And the influence of normal pressure on the cycle response of the new 3D-printed TPMS bridge bearing was studied by numerical simulation.

This is an interesting study that concerns a relatively specialized research topic. I would like to offer the following comments:

1. Page 2 (line 54-56): Authors mentioned that “the installation approaches of bridge bearings in bridge system around the world can be found as fully bonded, fully unbonded, and single-side bonded. In this paper, the rest is not considered for the cyclic response of a novel TPMS bridge bearing.”  There are three installation approaches of bridge bearings. However, in Chapter 2, only the fully bonded TPMS bridge bearing was introduced. Whether the other two cases are the same? the authors need to explain in detail.

2. Page 3 (line 136-137): “Noting that the bolt fracture damages are not considered in this study.” The failure of the bolt will lead to the failure of the bearing, is it reasonable not to consider the fracture damage of bolts?

3. Page 6: The simulation program shown in Table 1 includes monotone and cyclic programs, while Figure 6 only shows the displacement time history of cyclic loading, and the author should also give the displacement time history of monotone loading.

4. The latest publications associated the bridge seismic cyclic response should be cited in the paper. Zhong et al. investigated the hysteristic analysis of UBPRC columns of bridges (2022 a), and proposed novel models to assess the damage states of bridge columns (2022 b) . The investigated studies proved that ductility bridge columns would experience damage in extreme earthquake, therefore isolation bearing is an alternative method to reduce the seismic demand.  

Zhong, Jian; Shi, Longfei, Yang, Tao; Liu, Xiaoxian*; Wang, Yixian. Probabilistic seismic demand model of UBPRC columns conditioned on Pulse-Structure parameters. Engineering Structures. 2022 (a),270:114829.

Zhong, Jian; Ni, Ming; Hu, Huiming; Yuan, Wancheng; Yuan, Haiping; Pang, Yutao*; Uncoupled multivariate power models for estimating performance-based seismic damage states of column curvature ductility; Structures, 2022 (b), 36: 752-764.

5. Page 7: Formula 2 should be illustrated with an appropriate figure. The reader cannot clearly find the value of d1, d2, F1 and F2 let alone the points 1 and 2 mentioned in lines 254-255.

6. Page 10 (line 315-316): “These markers are employed to categories into the three phases of the proposed model responses, as previously indicated in Figures 8 and 9 above.” The three phases cannot be clearly shown from Figure 8 and Figure 9. In addition, the author should give a detailed description of the definition and description of the three phases.

Author Response

Dear reviewer,

Best regards,

Pasakorn

Reviewer 2 Report

The paper titled "Investigations into the effects of normal pressures on dynamic cyclic responses of a novel 3D-printed TPMS bridge bearing" deals with an innovative type of bridge bearings built using the 3D-printing technique adopting triply periodic minimal surface (TPMS).

The topic is interesting and worth of investigation, but the research has many aspects that require improvements before the draft can be accepted for publication.

     * Improve the introduction section.

“Bearings, also well known as isolators” (raw 26) bridge isolators and bearings are not exactly the same, so, if you intend to deal with isolators, you should make it evident from the beginning, starting with the wording in the title.

The seismic action, that justifies the adoption of isolators, is not mentioned through all the manuscript. This aspect puzzles the reader. Please clarify!

2.      * Explain in details the 3D-printing techniques adopted and TPMS. Paragraph 3.1 should become a separate section.

3.       * Explain better the type of simulations you are performing and the experimental data you are using for calibrating your results.

4.       * Improve paper readability.

Author Response

Dear reviewer,

Best regards,

Pasakorn

Round 2

Reviewer 2 Report

The authors have complyed with the reviewer requests.